# REPRESENTING UNORDERED DATA USING MULTISET AUTOMATA AND COMPLEX NUMBERS

## ABSTRACT

Unordered, variable-sized inputs arise in many settings across multiple fields. The ability for set- and multiset- oriented neural networks to handle this type of input has been the focus of much work in recent years. We propose to represent multisets using complex-weighted *multiset automata* and show how the multiset representations of certain existing neural architectures can be viewed as special cases of ours. Namely, (1) we provide a new theoretical and intuitive justification for the Transformer model's representation of positions using sinusoidal functions, and (2) we extend the DeepSets model to use complex numbers, enabling it to outperform the existing model on an extension of one of their tasks.

## 1 INTRODUCTION

Neural networks which operate on set-structured input have been gaining interest for their ability to handle unordered and variable-sized inputs (Vinyals et al., 2015; Wagstaff et al., 2019). They have been applied to various tasks, such as processing graph nodes (Murphy et al., 2018), hypergraphs (Maron et al., 2019), 3D image reconstruction (Yang et al., 2019), and point cloud classification and image tagging (Zaheer et al., 2017). Similar network structures have been applied to multiple instance learning (Pevný and Somol, 2016).

In particular, the DeepSets model (Zaheer et al., 2017) computes a representation of each element of the set, then combines the representations using a commutative function (e.g., addition) to form a representation of the set that discards ordering information. Zaheer et al. (2017) provide a proof that any function on sets can be modeled this way, by encoding sets as base-4 fractions and using the universal function approximation theorem, but their actual proposed model is far simpler than the model constructed by the theorem.

In this paper, we propose to compute representations of multisets using *weighted multiset automata*, a variant of weighted finite-state (string) automata in which the order of the input symbols does not affect the output. In some sense, this is the most general representation of a multiset that can be computed incrementally using only a finite amount of memory, and it can be directly implemented inside a neural network. We show how to train these automata efficiently by approximating them with string automata whose weights form complex, diagonal matrices.

Our representation generalizes DeepSets slightly, and it also turns out to be a generalization of the Transformer's position encodings (Vaswani et al., 2017). In Sections 4 and 5, we discuss the application of our representation in both cases.

- The Transformer (Vaswani et al., 2017) models the absolute position of a word within a sentence. This position can be thought of as a multiset over a single element, and indeed the Transformer uses a position encoding involving sinusoidal functions that turns out to be a special case of our representation. So weighted multiset automata provide a new theoretical and intuitive justification for sinusoidal position encodings. We also experiment with several variations on position encodings

inspired by this justification, and although they do not yield any improvement, we do find that learned position encodings in our representation do better than learning a different vector for each absolute position.

- We extend the DeepSets model to use our representation, which amounts to upgrading it from real to complex numbers. On an extension of one of their tasks (adding a sequence of one-digit numbers and predicting the units digit), our model is able to reach perfect performance, whereas the original DeepSets model does no better than chance.

## 2 WEIGHTED MULTISET AUTOMATA

We define weighted finite automata below using a matrix formulation. Throughout, let $\mathbb{K}$ be either $\mathbb{R}$ or $\mathbb{C}$.

**Definition 1.** *A $\mathbb{K}$-weighted finite automaton (WFA) over $\Sigma$ is a tuple $M = (Q, \Sigma, \lambda, \mu, \rho)$, where $Q = \{1, \ldots, d\}$ is a finite set of states, $\Sigma$ is a finite alphabet, $\lambda \in \mathbb{K}^{1 \times d}$ is a row vector of initial weights, $\mu : \Sigma \to \mathbb{K}^{d \times d}$ assigns a transition matrix to every symbol, and $\rho \in \mathbb{K}^{d \times 1}$ is a column vector of final weights.*

(We do not use the final weights $\rho$ in this paper, but include them for completeness.)

We extend the mapping $\mu$ to strings: If $w = w_1 \cdots w_n \in \Sigma^*$, then $\mu(w) = \mu(w_1) \cdots \mu(w_n)$. Then, the vector of *forward weights* of a string $w$ is $\text{fw}_M(w) = \lambda \left( \prod_{p=1}^n \mu(w_p) \right)$.

Note that, different from many definitions of weighted automata, this definition does not allow $\epsilon$-transitions, and there may be more than one initial state. (Throughout this paper, we use $\epsilon$ to stand for a small real number.)

The analogue of finite automata for multisets is the special case of the above definition where multiplication of the transition matrices $\mu(a)$ does not depend on their order.

**Definition 2.** *A $\mathbb{K}$-weighted multiset finite automaton is one whose transition matrices commute pairwise. That is, for all $a, b \in \Sigma$, we have $\mu(a)\mu(b) = \mu(b)\mu(a)$.*

Our proposal, then, is to represent a multiset $w$ by the vector of forward weights, $\text{fw}_M(w)$, with respect to some weighted multiset automaton $M$. In the context of a neural network, the transition weights $\mu(a)$ can be computed by any function as long as it does not depend on the ordering of symbols, and the forward weights can be used by the network in any way whatsoever.

## 3 TRAINING

Definition 2 does not lend itself well to training, because parameter optimization needs to be done subject to the commutativity constraint. Previous work (DeBenedetto and Chiang, 2018) suggested approximating training of a multiset automaton by training a string automaton while using a regularizer to encourage the weight matrices to be close to commuting. However, this strategy cannot make them commute exactly, and the regularizer, which has $O(|\Sigma|^2)$ terms, is expensive to compute.

Here, we pursue a different strategy, which is to restrict the transition matrices $\mu(a)$ to be diagonal. This guarantees that they commute. As a bonus, diagonal matrices are computionally less expensive than full matrices. Furthermore, we show that if we allow complex weights, we can learn multisets with diagonal matrices almost as well as with full matrices. We show this first for the special case of unary automata (§3.1) and then general multiset automata (§3.2).

## 3.1 UNARY AUTOMATA

Call an automaton *unary* if $|\Sigma| = 1$. Then, for brevity, we simply write $\mu$ instead of $\mu(a)$ where $a$ is the only symbol in $\Sigma$.

Let $\| \cdot \|$ be the Frobenius norm; by equivalence of norms (Horn and Johnson, 2012, 352), the results below should carry over to any other matrix norm, as long as it is monotone, that is: if $A \leq B$ elementwise, then $\|A\| \leq \|B\|$.

As stated above, our strategy for training a unary automaton is to allow $\mu$ to be complex but restrict it to be diagonal. The restriction does not lose much generality, because any matrix can approximated by a complex diagonal matrix in the following sense (Horn and Johnson, 2012, 116):

**Proposition 1.** *For any complex square matrix $A$ and $\epsilon > 0$, there is a complex matrix $E$ such that $\|E\| \leq \epsilon$ and $A + E$ is diagonalizable in $\mathbb{C}$.*

*Proof.* Form the Jordan decomposition $A = PJP^{-1}$. We can choose a diagonal matrix $D$ such that $\|D\| \leq \frac{\epsilon}{\kappa(P)}$ (where $\kappa(P) = \|P\|\|P^{-1}\|$) and the diagonal entries of $J + D$ are all different. Then $J + D$ is diagonalizable. Let $E = PDP^{-1}$; then $\|E\| \leq \|P\|\|D\|\|P^{-1}\| = \kappa(P)\|D\| \leq \epsilon$, and $A + E = P(J + D)P^{-1}$ is also diagonalizable. $\square$

Thus, for a unary automaton $M$ with transition matrix $\mu$, we can choose $Q\mu'Q^{-1}$ close to $\mu$ such that $\mu'$ is diagonal. So $M$ is close to the automaton with initial weights $\lambda' = \lambda Q$ and transition weights $\mu' \approx Q^{-1}\mu Q$. This means that in training, we can directly learn complex initial weights $\lambda'$ and a complex diagonal transition matrix $\mu'$, and the resulting automaton $(M')$ should be able to represent multisets almost as well as a general unary automaton $(M)$ can.

It might be thought that even if $\mu'$ approximates $\mu$ well, perhaps the forward weights, which involve possibly large powers of $\mu$, will not be approximated well. As some additional assurance, we have the following error bound on the powers of $\mu$:

**Proposition 2.** *For any complex square matrix $A$, $\epsilon > 0$, and $0 < r < 1$, there is a complex matrix $E$ such that $A + E$ is diagonalizable in $\mathbb{C}$ and, for all $n \geq 0$,*

$$\|(A+E)^n - A^n\| \leq r^n \epsilon \qquad \text{if $A$ nilpotent,}$$

$$\frac{\|(A+E)^n - A^n\|}{\|A^n\|} \leq n\epsilon \qquad \text{otherwise.}$$

For the proof, please see Appendix A.

## 3.2 GENERAL CASE

In this section, we allow $\Sigma$ to be of any size. Proposition 1 unfortunately does not hold in general for multiple matrices (O'Meara and Vinsonhaler, 2006). That is, it may not be possible to perturb a set of commuting matrices so that they are *simultaneously* diagonalizable.

**Definition 3.** *Matrices $A_1, \ldots, A_m$ are* simultaneously diagonalizable *if there exists an invertible matrix $P$ such that $PA_iP^{-1}$ is diagonal for all $i \in \{1, \cdots, n\}$.*

*We say that $A_1, \cdots, A_m$ are* approximately simultaneously diagonalizable (ASD) *if, for any $\epsilon > 0$, there are matrices $E_1, \ldots, E_m$ such that $\|E_i\| \leq \epsilon$ and $A_1 + E_1, \ldots, A_m + E_m$ are simultaneously diagonalizable.*

O'Meara and Vinsonhaler (2006) give examples of sets of matrices that are commuting but not ASD. However, if we are willing to add new states to the automaton (that is, to increase the dimensionality of the weight matrices), we can make them ASD.

**Proposition 3.** *Any weighted multiset automaton is close to an automaton that can be converted to a complex-weighted diagonal automaton, possibly with more states.*

*Proof.* First we start with a fact from O'Meara and Vinsonhaler (2006).

**Lemma 4.** *Suppose $A_1, \ldots, A_k$ are commuting $n \times n$ matrices over an algebraically closed field $F$. Then there exists an invertible matrix $C$ such that $C^{-1}A_1C, \ldots, C^{-1}A_kC$ are block diagonal matrices with matching block structures and each diagonal block has only a single eigenvalue (ignoring multiplicities). That is, there is a partition $n = n_1 + \cdots + n_r$ of $n$ such that*

$$C^{-1}A_iC = B_i = \begin{bmatrix} B_{i1} & & & \\ & B_{i2} & & \\ & & \ddots & \\ & & & B_{ir} \end{bmatrix}, \tag{1}$$

*where each $B_{ij}$ is an $n_j \times n_j$ matrix having only a single eigenvalue for $i = 1, \ldots, k$ and $j = 1, \ldots, r$. Moreover, if $B_{1j}, \ldots, B_{kj}$ are ASD for $j = 1, \ldots, r$, then $A_1, \ldots, A_k$ are ASD.*

Furthermore, O'Meara and Vinsonhaler (2006) observe that each block can be written as $B_{ij} = \lambda_{ij}I + N_{ij}$ where $N_{ij}$ is nilpotent, so $A_1, \ldots, A_k$ are ASD iff $N_{1j}, \ldots, N_{kj}$ are for all $j$.

So the transition matrices of the automaton can be rewritten in the above form, and the problem of converting an automaton to one that is ASD is reduced to the problem of converting an automaton with nilpotent transition matrices (equivalently, an automaton recognizing a finite language) to one that is ASD (possibly with more states). See Appendix B for one such construction. □

This means that if we want to learn representations of multisets over a finite alphabet $\Sigma$, it suffices to constrain the transition matrices to be complex diagonal, possibly with more states. Unfortunately, the best construction we know of (Appendix B) increases the number of states by a lot. But this does not in any way prevent the use our representation; we can choose however many states we want, and it's an empirical question whether the number of states is enough to learn good representations.

The following two sections look at two practical applications of our representation.

## 4 POSITION ENCODINGS

One of the distinguishing features of the Transformer network for machine translation (Vaswani et al., 2017), compared with older RNN-based models, is its curious-looking *position encodings*,

$$\begin{aligned} \mathbf{e}_{2j-1}^p &= \sin 10000^{-2(j-1)/d}(p-1) \\ \mathbf{e}_{2j}^p &= \cos 10000^{-2(j-1)/d}(p-1) \end{aligned} \tag{2}$$

which map word positions $p$ (ranging from 1 to $n$, the sentence length) to points in the plane and are the model's sole source of information about word order.

In this section, we show how these position encodings can be interpreted as the forward weights of a weighted unary automaton. We also report on some experiments on some extensions of position encodings inspired by this interpretation.

### 4.1 AS A WEIGHTED UNARY AUTOMATON

Consider a diagonal unary automaton $M$ in the following form:

$$\lambda = \begin{bmatrix} \exp i\phi_1 & \exp -i\phi_1 & \exp i\phi_2 & \exp -i\phi_2 & \cdots \end{bmatrix}$$

$$\mu = \begin{bmatrix} \exp i\theta_1 & 0 & 0 & 0 & \cdots \\ 0 & \exp -i\theta_1 & 0 & 0 & \cdots \\ 0 & 0 & \exp i\theta_2 & 0 & \cdots \\ 0 & 0 & 0 & \exp -i\theta_2 & \cdots \\ \vdots & \vdots & \vdots & \vdots & \ddots \end{bmatrix}$$

In order for a complex-weighted automaton to be equivalent to some real-weighted automaton, the entries must come in conjugate pairs like this, so this form is fully general.

By a change of basis, this becomes the following unary automaton $M'$ (this is sometimes called the real Jordan form):

$$\lambda' = \begin{bmatrix} \cos \phi_1 & \sin \phi_1 & \cos \phi_2 & \sin \phi_2 & \cdots \end{bmatrix}$$

$$\mu' = \begin{bmatrix} \cos \theta_1 & \sin \theta_1 & 0 & 0 & \cdots \\ -\sin \theta_1 & \cos \theta_1 & 0 & 0 & \cdots \\ 0 & 0 & \cos \theta_2 & \sin \theta_2 & \cdots \\ 0 & 0 & -\sin \theta_2 & \cos \theta_2 & \cdots \\ \vdots & \vdots & \vdots & \vdots & \ddots \end{bmatrix} \tag{3}$$

Then, for any string prefix $u$ (making use of the angle sum identities):

$$\mathrm{fw}_{M'}(u) = \begin{bmatrix} \cos(\phi_1 + |u|\theta_1) & \sin(\phi_1 + |u|\theta_1) & \cos(\phi_2 + |u|\theta_2) & \sin(\phi_2 + |u|\theta_2) & \cdots \end{bmatrix}.$$

If we let

$$\phi_i = \frac{\pi}{2}$$
$$\theta_j = -10000^{-2(j-1)/d}$$

this becomes exactly equal to the position encodings defined in (2). Thus, the Transformer's position encodings can be reinterpreted as follows: it runs automaton $M'$ over the input string and uses the forward weights of $M'$ just before position $p$ to represent $p$. This encoding, together with the embedding of word $w_p$, is used as the input to the first self-attention layer.

### 4.2 EXPERIMENTS

This reinterpretation suggests some potential extensions to position encodings: 1. Using the diagonal, polar form of the transition matrix (3), learn the $\phi_i$ and $\theta_i$ instead of keeping them fixed. 2. Learn all the initial weights and full transition matrix directly.

We carried out some experiments to see if these methods perform better or worse than the original. We used an open-source implementation of the Transformer, Witwicky.[1] The settings used were the default settings, except that we used 8k joint BPE operations and $d = 512$ embedding dimensions. We tested the following variations on position encodings.

---

[1] https://github.com/tnq177/witwicky

| Model | Training | En-Vi* | Uz-En | case-insensitive BLEU | | | | |
| | | | | Ha-En | Hu-En | Ur-En | Ta-En | Tu-En |
| --- | --- | --- | --- | --- | --- | --- | --- | --- |
| diagonal polar | fixed | 32.6 | 25.7 | 24.4 | **34.2** | 11.5 | 13.4 | 25.7 |
| | learned angles | **32.7** | 25.8 | 25.4 | 34.0 | 11.1 | 14.1 | 25.7 |
| full matrix | random | 32.6 | **25.9** | **25.8** | 34.1 | 10.9 | 12.8 | **26.1** |
| | learned | 32.5 | 24.5 | 23.6 | 33.5 | 11.4 | **14.5** | 23.8 |
| per position | random | 32.6 | 24.9 | 24.6 | 34.1 | 11.0 | 14.1 | 24.4 |
| | learned | 32.1 | 22.6 | 21.2 | 33.0 | **11.7** | 14.4 | 21.1 |

*tokenized references

Table 1: Machine translation experiments with various position encodings. Scores are in case-insensitive BLEU, a common machine translation metric. The best score in each column is printed in boldface.

- Diagonal polar
  - fixed: The original sinusoidal encodings (Vaswani et al., 2017).
  - learned angles: Initialize the $\phi_i$ and $\theta_i$ to the original values, then optimize them (#1 above).
- Full matrix
  - random: Randomize initial weights so that their expected norm is the same as the original, and transition matrix using orthogonal initialization (Saxe et al., 2013), and do not optimize them.
  - learned: Initialize $\lambda$ and $\mu$ as above, and then optimize them (#2 above).
- Per position
  - random: Choose a vector with fixed norm and random angle for each absolute position, and do not optimize them.
  - learned: Learn an encoding for each absolute position (Gehring et al., 2017).

Table 1 shows that no method is clearly the best. The only method that appears to be worse than the others is "per position, learned," which, although best on Urdu-English, does much worse than other methods on several tasks. By contrast, the learned embeddings based on multiset automata ("diagonal polar, learned angles" and "full matrix, learned") are usually close to the best, lending some support to our interpretation.

## 5 COMPLEX DEEPSETS

In this sectoin, we incorporate a weighted multiset automaton into the DeepSets (Zaheer et al., 2017) model, extending it to use complex numbers.

### 5.1 MODELS

The DeepSets model computes a vector representation for each input symbol and sums them to discard ordering information. We may think of the elementwise layers as computing the log-weights of a diagonal multiset automaton, and the summation layer as computing the forward log-weights of the multiset. (The logs are needed because DeepSets adds, whereas multiset automata multiply.) However, DeepSets uses only real weights, whereas our multiset automata use complex weights. Thus, DeepSets can be viewed as using a multiset representation which is a special case of ours.

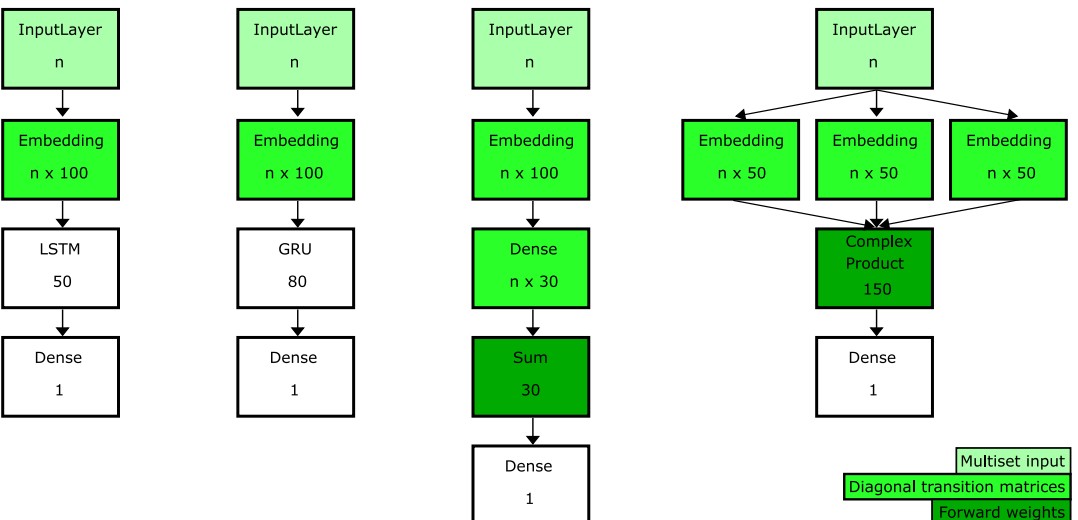

Figure 1: Neural architecture for (left to right) LSTM, GRU, DeepSets, and our model. Each cell indicates layer type and output dimension. Color indicates which part of a multiset automaton it corresponds to, though only our model is fully general for multiset automata.

We conduct experiments comparing the DeepSets model (Zaheer et al., 2017), a GRU model, an LSTM model, and our complex multiset model. The code and layer sizes for the three baselines come from the DeepSets paper.[2] See Figure 1 for layer types and sizes for the three baseline models.

In our system, to avoid underflow when multiplying many complex numbers, we store each complex number as $e^r(a+bi)$ where $r$, $a$, and $b$ are real and $a$ and $b$ are normalized such that $a^2+b^2 = 1$ prior to multiplication. Thus, for each complex-valued parameter, we have three scalars ($r$, $a$, and $b$) to learn. To this end, each input is fed into three separate embedding layers of size 50 (for $r$, $a$, and $b$). (While the DeepSets code uses a dense layer at this point, in our network, we found that we could feed the embeddings directly into a complex multiplication layer to discard ordering information. This reduced the number of parameters for our model and did not affect performance.) The output of this is then a new $r$, $a$, and $b$ which are concatenated and fed into a final dense layer as before to obtain the output. Since our diagonalized automata have complex initial weights ($\lambda'$), we also tried learning a complex initial weight vector $\lambda'$, but this had no effect on performance.

The total number of parameters for each model was 4,161 parameters for the DeepSets model, 31,351 parameters for the LSTM model, 44,621 parameters for the GRU model, and 1,801 parameters for our model. In order to eliminate number of parameters as a difference from our model to the DeepSets model, we also tried the DeepSets model without the first dense layer and with embedding sizes of 150 to exactly match the number of parameters of our model, and the results on the test tasks were not significantly different from the baseline DeepSets model.

For all experiments, we used mean squared error loss, a learning rate decay of 0.5 after the validation loss does not decrease for 2 epochs, and early stopping after the validation loss does not decrease for 10 epochs.

---

[2]https://github.com/manzilzaheer/DeepSets/blob/master/DigitSum/text_sum.ipynb

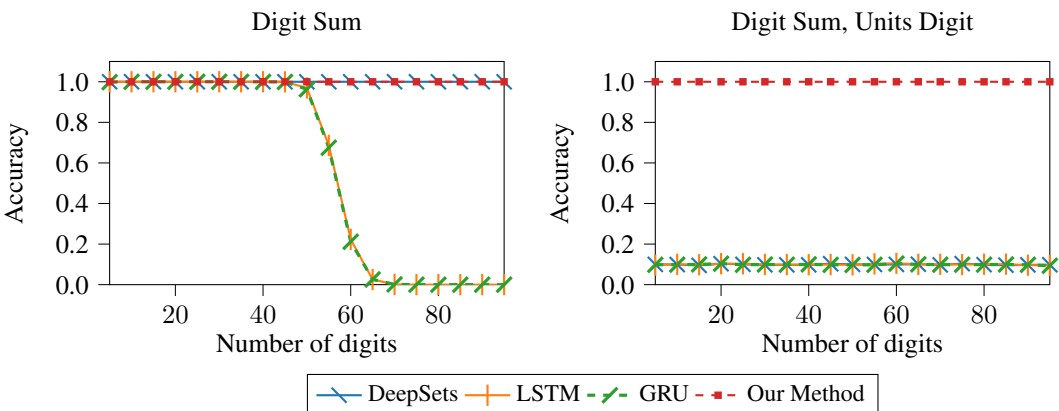

Figure 2: Results for task 1 (left) and task 2 (right). In task 1, the LSTM and GRU models were unable to generalize to examples larger than seen in training, while DeepSets and our model generalize to all test lengths. For task 2, only our model is able to return the correct units digit for all test lengths. The GRU, LSTM, and DeepSets models fail to learn any behavior beyond random guessing.

### 5.2 EXPERIMENTS

**Task 1: Sum of digits**  In this task, taken from Zaheer et al. (2017), the network receives a set of single digit integers as input and must output the sum of those digits. The output is rounded to the nearest integer to measure accuracy. The training set consisted of 100k randomly generated sequences of digits 1–9 with lengths from 1 to 50. They were fed to each network in the order in which they were generated (which only affects GRU and LSTM). This was then split into training and dev with approximately a 99/1 split. The test set consisted of randomly generated sequences of lengths that were multiples of 5 from 5 to 95. Figure 2 shows that both our model and DeepSets obtain perfect accuracy on the test data, while the LSTM and GRU fail to generalize to longer sequences.

**Task 2: Returning units digit of a sum**  The second task is similar to the first, but only requires returning the units digit of the sum. The data and evaluation are otherwise the same as task 1. Here, random guessing within the output range 0–9 achieves approximately 10% accuracy. Figure 2 show that DeepSets, LSTM, and GRU are unable to achieve performance better than random guessing on the test data. Our method is able to return the units digit perfectly for all test lengths, because it effectively learns to use the cyclic nature of complex multiplication to produce the units digit.

## 6 CONCLUSION

We have proven that weighted multiset automata can be approximated by automata with (complex) diagonal transition matrices. This formulation permits simpler elementwise multiplication instead of matrix multiplication, and requires fewer parameters when using the same number of states. We show that this type of automaton naturally arises within existing neural architectures, and that this representation generalizes two existing multiset representations, the Transformer's position encodings and DeepSets. Our results provide new theoretical and intuitive justification for these models, and, in one case, lead to a change in the model that drastically improves its performance.

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

## A PROOF OF PROPOSITION 2

**Lemma 5.** *If $J$ is a Jordan block with nonzero eigenvalue, then the bound of Proposition 2 holds for $J$.*

*Proof.* Note that for any $\delta, \epsilon \geq 0$, we have

$$(1 - \delta)(1 - \epsilon) \geq 1 - \delta - \epsilon \geq 1 - 2\max\{\delta, \epsilon\}$$
$$(1 - \epsilon)^n \geq 1 - n\epsilon$$

The powers of $J$ look like

$$J^n = \begin{bmatrix} \binom{n}{0}\lambda^n & \binom{n}{1}\lambda^{n-1} & \binom{n}{2}\lambda^{n-2} & \cdots \\ & \binom{n}{0}\lambda^n & \binom{n}{1}\lambda^{n-1} & \cdots \\ & & \binom{n}{0}\lambda^n & \cdots \\ & & & \ddots \end{bmatrix}$$

More concisely, for $k \geq j$,

$$[J^n]_{jk} = \binom{n}{k-j}\lambda^{n-k+j}.$$

Let $D$ be a diagonal matrix whose elements are in $[-\epsilon\lambda, 0)$ and are all different. The powers of $(J + D)$ are

$$[(J + D)^n]_{jk} = c_{jk}[J^n]_{jk}$$

where

$$\begin{aligned} c_{jk} &\geq (1 - \epsilon)^{n-k+j} \\ &\geq 1 - (n - k + j)\epsilon \\ &\geq 1 - n\epsilon. \end{aligned}$$

Finally, form their difference:

$$\begin{aligned} [(J + D)^n - J^n]_{jk} &= (c_{jk} - 1)[J^n]_{jk} \\ |[(J + D)^n - J^n]_{jk}| &\leq n\epsilon \, |[J^n]_{jk}| \\ \|(J + D)^n - J^n\| &\leq n\epsilon \, \|J^n\| \\ \frac{\|(J + D)^n - J^n\|}{\|J^n\|} &\leq n\epsilon. \end{aligned}$$

$\square$

**Lemma 6.** *If $J$ is a Jordan block with zero eigenvalue, then for any $\epsilon > 0, r > 0$, there is a complex matrix $E$ such that $M + E$ is diagonalizable in $\mathbb{C}$ and*

$$\|(M + E)^n - M^n\| \leq r^n \epsilon.$$

*Proof.* In this case, we have to perturb the diagonal elements to nonzero values. For any $\delta \leq \frac{1}{2}$, let $D$ be a diagonal matrix whose elements are in $(0, \delta]$ and are all different. Then the elements of $((J + D)^n - J^n)$ are:

$$\begin{aligned} [(J + D)^n - J^n]_{jk} &\leq \binom{n}{k-j}\delta^{n-k+j} \qquad (0 \leq k - j < \min\{n, d\}) \\ &< 2^n \delta^{\min\{0, n-d\}+1} \end{aligned}$$

so the error is at most $2^{d-1}d(2\delta)^{\min\{0,n-d\}+1}$. Let $\delta = \min\{\frac{1}{2}, r, \left(\frac{r}{2}\right)^d \frac{\epsilon}{d}\}$. $\square$

Now we can prove Proposition 2.

*Proof.* Form the Jordan decomposition $M = PJP^{-1}$, where

$$
J = \begin{bmatrix} J_1 & & & \\ & J_2 & & \\ & & \ddots & \\ & & & J_p \end{bmatrix}.
$$

We begin with the non-nilpotent case. Let $\kappa(P) = \|P\|\|P^{-1}\|$ be the Frobenius condition number of $P$. For each Jordan block $J_j$:

- If $J_j$ has nonzero eigenvalue, perturb it so that the absolute error of the $n$th power is at most $\frac{n\epsilon}{\kappa(P)^2} \frac{\|J_j^n\|}{2p}$, by Lemma 5.

- If $J_j$ has zero eigenvalue, perturb it so that the absolute error is at most $\frac{n\epsilon}{\kappa(P)^2} \frac{\rho(J)^n}{2p}$, by Lemma 6.

Then the total absolute error of all the blocks with nonzero eigenvalue is at most $\frac{n\epsilon}{\kappa(P)^2} \frac{\|J^n\|}{2}$. And since $\rho(J)^n \le \|J^n\|$, the total absolute error of all the blocks with zero eigenvalue is also at most $\frac{n\epsilon}{\kappa(P)^2} \frac{\|J^n\|}{2}$. So the combined total is

$$
\|(J + E)^n - J^n\| \le \frac{n\epsilon}{\kappa(P)^2} \|J^n\|.
$$

Finally,

$$
\begin{aligned}
\|(M + E)^n - M^n\| &= \|P((J + D)^n - J^n)P^{-1}\| \\
&\le \kappa(P)\|((J + D)^n - J^n)\| \\
&\le \frac{n\epsilon}{\kappa(P)}\|J^n\| \\
&\le \frac{n\epsilon}{\kappa(P)}\|P^{-1}M^n P\| \\
&\le n\epsilon\|M^n\| \\
\frac{\|(M + E)^n - M^n\|}{\|M^n\|} &\le n\epsilon.
\end{aligned}
$$

$\square$

If $M$ is nilpotent, the above argument does not go through, because $\rho(J) = 0$. Instead, use Lemma 6 to bound the absolute error of each block by $\frac{r^n \epsilon}{p}$, so that the total absolute error is at most $r^n \epsilon$.

## B  MAKING AUTOMATA ASD

In this section, we give a construction for converting a multiset automaton to one that is equivalent, but possibly has more states.

Let $\oplus$ stand for the direct product of vector spaces, $\otimes$ for the Kronecker product, and define the shuffle product $A \sqcup B = A \otimes I + I \otimes B$. (This is known as the Kronecker sum and is sometimes notated $\oplus$, but we

use that for direct product.) These operations extend naturally to weighted multiset automata and correspond roughly to union and concatenation, respectively:

$$\lambda_{A \oplus B} = \lambda_A \oplus \lambda_B \qquad \mu_{A \oplus B}(a) = \mu_A(a) \oplus \mu_B(a) \qquad \rho_{A \oplus B} = \rho_A \oplus \rho_B$$
$$\lambda_{A \sqcup B} = \lambda_A \otimes \lambda_B \qquad \mu_{A \sqcup B}(a) = \mu_A(a) \sqcup \mu_B(a) \qquad \rho_{A \sqcup B} = \rho_B \otimes \rho_B$$

They are of interest here because they preserve the ASD property:

**Proposition 7.** *If $M_1$ and $M_2$ are multiset automata with ASD transition matrices, then $M_1 \oplus M_2$ has ASD transition matrices, and $M_1 \sqcup M_2$ has ASD transition matrices.*

*Proof.* First consider the $\oplus$ operation. Let $\mu_1(a)$ (for all $a$) be the transition matrices of $M_1$. For any $\epsilon > 0$, let $E_1(a)$ be the perturbations of the $\mu_1(a)$ such that $\|E_1(a)\| \le \epsilon/2$ and the $\mu_1(a) + E_1(a)$ (for all $a$) are simultaneously diagonalizable. Similarly for $M_2$. Then the matrices $(\mu_1(a) + E_1(a)) \oplus (\mu_2(a) + E_2(a))$ (for all $a$) are simultaneously diagonalizable, and

$$\|(\mu_1(a) + E_1(a)) \oplus (\mu_2(a) + E_2(a)) - \mu_1(a) \oplus \mu_2(a)\| = \|E_1(a) \oplus E_2(a)\|$$
$$\le \|E_1(a)\| + \|E_2(a)\|$$
$$= \epsilon.$$

Next, we consider the $\sqcup$ operation. Let $d_1$ and $d_2$ be the number of states in $M_1$ and $M_2$, respectively. This time, we choose $\|E_1(a)\| \le \epsilon/(2d_2)$ and $\|E_2(a)\| \le \epsilon/(2d_1)$. Then the matrices $(\mu_1(a) + E_1(a)) \sqcup (\mu_2(a) + E_2(a))$ (for all $a$) are simultaneously diagonalizable, and

$$(\mu_1(a) + E_1(a)) \sqcup (\mu_2(a) + E_2(a)) = (\mu_1(a) + E_1(a)) \otimes I + I \otimes (\mu_2(a) + E_2(a))$$
$$= \mu_1(a) \otimes I + E_1(a) \otimes I + I \otimes \mu_2(a) + I \otimes E_2(a)$$
$$= (\mu_1(a) \sqcup \mu_2(a)) + (E_1(a) \sqcup E_2(a))$$
$$\|(\mu_1(a) + E_1(a)) \sqcup (\mu_2(a) + E_2(a)) - \mu_1(a) \sqcup \mu_2(a)\| = \|E_1(a) \sqcup E_2(a)\|$$
$$= \|E_1(a) \otimes I + I \otimes E_2(a)\|$$
$$\le \|E_1(a) \otimes I\| + \|I \otimes E_2(a)\|$$
$$\le \|E_1(a)\|d_2 + d_1\|E_2(a)\|$$
$$\le \epsilon.$$

$\square$

**Proposition 8.** *If $M$ is a weighted multiset automaton with $d$ states recognizing a finite language (that is, all of its transition matrices are nilpotent), there exists an equivalent automaton with $O(d^{2|\Sigma|+1})$ states whose transition matrices are ASD.*

*Proof.* Because any set of commuting matrices can be simultaneously triangularized by a change of basis, assume without loss of generality that $M$'s transition matrices are upper triangular, that is, there are no transitions from state $q$ to state $r$ where $q > r$.

The idea is that $M'$ should simulate a run of $M$ in which the symbols are read in lexicographic order. It does so by building up partial runs, one for each symbol in $\Sigma$, and then stitching them together.

Let $Q$ be the states of $M$, and let $a_1, \dots, a_m$ be the symbols of $\Sigma$. For all $a \in \Sigma, q, r \in Q$, define $M_{q,a,r}$ to be the automaton which simulates $M$ starting from state $q$, reading only $a$'s, and ending in state $r$. Then

$$M' = \bigoplus_{q_0 \in Q} \cdots \bigoplus_{q_m \in Q} \lambda_{q_0} M_{q_0, a_1, q_1} \sqcup \cdots \sqcup M_{q_{m-1}, a_m, q_m}$$

(where the multiplication by the scalar $\lambda_{q_0}$ means scaling the initial weight vector by $\lambda_{q_0}$). By Proposition 7, $M'$ is ASD, and because each of the $M_{q,a,r}$ has no more than $d$ states, $M'$ has at most $d^{2|\Sigma|+1}$ states. $\square$

