# OpenReview forum: "Representing Unordered Data Using Multiset Automata and Complex Numbers"
_ICLR.cc/2020/Conference — Reject_

### Official Review · AnonReviewer3 · 2019-10-22
**Official Blind Review #3**

**Rating:** 3

**Review:**

This work presents an encoding approach for unordered set input to neural networks. The authors base their approach on weighted finite automata, where in order to absorb unordered sets, they enforce multiplicative commutativity on transition matrices by approximating them as complex diagonal matrices. The authors furthermore provide mathematical references and results to derive bounds for their approximation. They show that positional encoding in Transformer network can be seen as a special case of their multiset encoding scheme, which also generalizes DeepSets encoding from real to complex numbers.

The paper is well-written and easy to follow. The work tries to unify positional encoding in Transformers and Deepsets by establishing a different view to multiset encoding. My major concern however is that the authors do not provide any reasoning as to why do we need the weighted automata machinery behind their approach? Effectively what they do can simply be seen as embedding of inherently periodic multiset elements using periodic functions, which are parameterized by non-linear transformations of data. Such encoding schemes have long been used in signal processing.

I might have missed something, but in my opinion the theoretical contribution of the work is rather tangential to the empirical analysis and results presented in the paper.

**Experience Assessment:**

I have read many papers in this area.

**Review Assessment: Checking Correctness Of Derivations And Theory:**

I did not assess the derivations or theory.

**Review Assessment: Checking Correctness Of Experiments:**

I carefully checked the experiments.

**Review Assessment: Thoroughness In Paper Reading:**

I read the paper at least twice and used my best judgement in assessing the paper.

---

> ### Author Response · Authors · 2019-11-07
> **Thank you for your review**
>
> Thank you for taking the time to read and review our paper.  We appreciate your feedback and below address your major concern regarding connections to signal processing.
>
> > My major concern however is that the authors do not provide any reasoning as to why do we need the weighted automata machinery behind their approach?
>
> There are some connections with signal processing as you mention, however in our system the input and output are not necessarily periodic in our setting.  For example, the digit summation problem (task 1) is not periodic in input or output and our system is able to handle this task as effectively as the original DeepSets network.  In task 2, the output has a periodic nature to it but the input again is not periodic.
> The signal processing connection is perhaps stronger in the Transformer case in which sines and cosines are used in the original encodings.  Here the original encoding is a specific form of weighted multiset automaton and it seemed natural in viewing it this way.  With this view in mind, we tried further generalizations that this suggests.  There may be other ways to view this encoding in a more traditional signal processing manner, but we do not explore that here.

---

### Official Review · AnonReviewer2 · 2019-10-23
**Official Blind Review #2**

**Rating:** 6

**Review:**

This paper proposed a complex weights based multiset automata designed to represent unordered data. The main idea of multiset automata is that the transition matrices of the automata is pairwise commutative. To achieve this property, the authors proposed to restrict the transition matrices to be diagonal and shows that the latter is a close approximation of the former. The authors proceed to give two practical applications of the multiset automata: position encoding of the transformer and deepset networks. For the former, the authors showed that the position encodings from Vaswani et al. can be written as a weighted unary automaton and therefore it is a generalization of the original position encodings. For the latter, the authors extended the classical deepset networks into its complex domain, allowing more efficient representation of the data.

I think this paper overall did a good job, and I really like the construction of the multiset automata and the theoretical guarantees the authors derived and the two applications are straight-forward to see. However I do find the motivation of this paper is a bit weak,  and I’m having a hard time finding the highlight of the paper. Therefore, I’m giving this paper a weak accept.

Here are some general comments:
How is the multiset automata learnt? For weighted automata, one classical way is to use spectral learning algorithm (see Balle et. al. 2014). In this paper, the learning aspect of the multiset automata was not mentioned. I assume that the authors use some kind of gradient descent to optimize the weights w.r.t the whole networks. However, I do think it’s important to let the readers know this key step.

For the first experiment on the position encoding. I really like the derivation here and it seems that theoretically, multiset automata should be a generalization of the position encodings. However, the experiments didn’t show much difference. Is there any potential explanation here? Moreover, what is the advantage of using multiset automata in transformers instead of the original position encoding? Is it the runtime is faster? Cause you only need to compute the diagonal parameters and thus drastically reduce the number of parameters? If so, a comparison of runtime might be useful here to further showcase the advantage of the multiset automata.

For the first application, it is great that the authors shows the connection, but it seems that it just stops at the level of showing the position encodings can be viewed as a multiset automata. For example, what happens if you don’t restrict it to be sinusoidal functions? e.g. set the transitions to be diagonal and directly optimize with gradient descent?
For the second experiment, I’m a little confused about the experiment setup and the baselines. First how do the authors incorporate multiset automata into the deepset models? In Figure 1, every neural architectures have this embedding layers with diagonal transition matrices, does this mean the multiset automata is applied to encode the input for every architecture? If so is it possible that the use of multiset automata in LSTM, GRU and deepset, is actually hurting the performance? Maybe a comparison with vanilla LSTM, GRU and deepset is also needed.

Moreover, for the second experiment, the size of LSTM, GRU, deepset seems to be smaller comparing to the complex product layer in the authors’ architecture (about half of the size). It is true that the authors mentioned that with non-unary automata, the size of the automata is significantly larger, but is this set-up fair for the baselines, e.g. if you use 150 size of LSTM, will it perform equally well to the multiset automata?

In addition, I have a bit of difficulty understanding why three embedding layers are needed to learn the complex number, is it possible to learn r, a, b jointly with one embedding?

Is there some real data experiment done for the second application (the extension to deepset), to further showcase the significance of using complex domain?

Here are some writing comments (did not affect the decision):
In page 6, the bullet point “Diagonal polar”, what does (#1 above) mean? Same goes for (#2 above) in the latter point.
In “Full matrix”, the sentence “…, and transition matrix using orthogonal initialization…” does not have a verb and is a bit confusing to read.

Page 7, second paragraph, line 4, in the brackets. There are two sentences in the brackets, and it feels a bit heavy and it actually says something important. Maybe either put it out or leave it as a footnote?

Overall, I like the concept of the multiset automata, but I feel there is a lack of highlights to further showcase this paper. I feel maybe a further investigation of either of these application could make a great paper.

**Experience Assessment:**

I have read many papers in this area.

**Review Assessment: Checking Correctness Of Derivations And Theory:**

I assessed the sensibility of the derivations and theory.

**Review Assessment: Checking Correctness Of Experiments:**

I assessed the sensibility of the experiments.

**Review Assessment: Thoroughness In Paper Reading:**

I read the paper at least twice and used my best judgement in assessing the paper.

---

> ### Author Response · Authors · 2019-11-07
> **Thank you for your review**
>
> Thank you for taking the time to read and review our paper.  We appreciate your feedback and below address questions you expressed.
>
> > How is the multiset automata learnt?
>
> We learn the automaton by gradient descent as part of training the larger network. It would certainly be interesting to think about spectral methods as well.
>
> > For the first experiment...Is there any potential explanation here? Moreover, what is the advantage of using multiset automata in transformers instead of the original position encoding?
>
> We think the explanation is simply that the "per position, learned" setting has more parameters and is more prone to overfitting. Our representation provides ways of parameterizing position encodings that generalize better. However, as long as overfitting is avoided, the Transformer seems fairly insensitive to the particular choice of position encoding.
>
> We consider this result primarily of conceptual interest, especially for people (like ourselves) who are initially baffled by the use of sinusoidal functions in the original position encoding.
>
> In terms of practical advantages, it's possible that in a situation where learned position encodings are needed, our parameterization might generalize better. For example, BERT uses learned, per-position encodings and some care is required to train them properly (train on sequences of length 128 for 90% of the steps, then sequences of length 512 for 10% of the steps). It's pure speculation, but maybe our parameterization would make this detail unnecessary.
>
> > For example, what happens if you don’t restrict it to be sinusoidal functions? e.g. set the transitions to be diagonal and directly optimize with gradient descent?
>
> If we set the transition matrix to be diagonal and complex, we have to choose a representation for complex numbers. If we use polar coordinates, the result is similar to the line in Table 1 labeled "learned angles." (This result was not shown, but could be included in a future version of the paper.)
> We did not try using rectangular coordinates, as for the units-digit experiment. That is certainly something that could be added to the paper, but we would not expect the results to be very different.
>
> > For the second experiment...First how do the authors incorporate multiset automata into the deepset models? In Figure 1, every neural architectures have this embedding layers with diagonal transition matrices, does this mean the multiset automata is applied to encode the input for every architecture? If so is it possible that the use of multiset automata in LSTM, GRU and deepset, is actually hurting the performance? Maybe a comparison with vanilla LSTM, GRU and deepset is also needed.
>
> Figure 1 is meant to show the corresponding similar structures in each architecture but does not indicate any change to the three baselines, I apologize if that was unclear.  The input to each network is a multiset since addition itself is commutative, but as noted in the description from task 1, they are fed to each network in the order in which they were generated.  This imposes an ordering for the GRU and LSTM whereas the input ordering information is discarded by DeepSets and our method.  The experiments are run on vanilla LSTM, GRU, and DeepSets networks, with figure 1 meant to show where there are similarities across the systems.
>
> (please see next reply for more)

---

> ### Author Response · Authors · 2019-11-07
> **Review response continued**
>
> > Moreover, for the second experiment, the size of LSTM, GRU, deepset seems to be smaller comparing to the complex product layer in the authors’ architecture (about half of the size). It is true that the authors mentioned that with non-unary automata, the size of the automata is significantly larger, but is this set-up fair for the baselines, e.g. if you use 150 size of LSTM, will it perform equally well to the multiset automata?
>
> The number of parameters for the LSTM and GRU networks is 31,351 and 44,621 respectively whereas in our method the number of parameters is 1,801. This is due to the fact that complex multiplication does not require any learned parameters, so only the embeddings and the final dense layer need to be learned.  By contrast, the LSTM and GRU layers themselves have many learned parameters.  Overall this results in more learned parameters required by the LSTM and GRU networks compared to ours.
>
> > In addition, I have a bit of difficulty understanding why three embedding layers are needed to learn the complex number, is it possible to learn r, a, b jointly with one embedding?
>
> Since each complex number is represented as $e^r(a+bi)$, there are three parts learned for each complex number.  This could be implemented as a single embedding of size $3n$ for some $n$, but this is effectively the same as concatenating the three smaller embeddings we used in our implementation.  We found it easier to keep the embeddings separate prior to the complex multiplication, but there is no difference in computational power.
>
> > Is there some real data experiment done for the second application (the extension to deepset), to further showcase the significance of using complex domain?
>
> The two tasks that were used were chosen to demonstrate the effectiveness of our method on handling the type of problem that DeepSets is well suited for (task 1) and a related type of problem that it struggles with (task 2).  This seemed like a good way to demonstrate, on a simple problem, one type of behavior that complex numbers are well suited to handle.  We do plan to utilize this type of network for more complicated, real world applications in the future.

---

### Official Review · AnonReviewer1 · 2019-10-23
**Official Blind Review #1**

**Rating:** 6

**Review:**

This paper proposes generating feature representations for set elements using weighted multiset automata. Experiments show that this leads to better generalization performance in some tasks.

I am leaning to reject this paper. The proposed algorithm for generating features seems relevant and correct, but there are shortcomings in the presentation and the experiments are not entirely convincing.

In particular, the paper begins by introducing weighted multiset automata quite clearly, but it fails to explain how exactly these automata would be used to generate set representations. I assumed that the set would be represented as the state of the automaton after processing a string (where each element of the set is a symbol from the alphabet in the string) but in section 4 the different states of the automaton while processing a string are used instead. If this paper proposes a new way of learning representations for sets, I would like to see a general recipe for the application of this idea.

Reading the paper it is not entirely clear what theoretical results are novel and which proofs are restatements of existing proofs. It would be useful to guide the reader a bit more clearly here.

The second statement in section 4.1 is not clear to me: In what sense is the diagonal with alternating complex conjugate entries fully general?

The experimental results are difficult to interpret. Since there are no confidence intervals it is impossible to draw conclusions from table 1. I am also not entirely convinced by figure 2. The "unit digit of a sum" task seems slightly artificially constructed to be suitable for a network which uses complex numbers. Although this is not a bad thing, it doesn't necessarily validate that complex weighted automata have better representational power. If that was the case, wouldn't we expect better results for other tasks that don't explicitly have a cyclic nature?

The main questions I would like to see answered (and adjusted in the paper) for me to accept this paper would be:

* What is the general recipe for applying this technique to get representations of a multiset?
* How do the experimental results validate the increased representational power of complex-weighted diagonal automata?

**Experience Assessment:**

I do not know much about this area.

**Review Assessment: Checking Correctness Of Derivations And Theory:**

I assessed the sensibility of the derivations and theory.

**Review Assessment: Checking Correctness Of Experiments:**

I assessed the sensibility of the experiments.

**Review Assessment: Thoroughness In Paper Reading:**

I read the paper at least twice and used my best judgement in assessing the paper.

---

> ### Author Response · Authors · 2019-11-07
> **Thank you for your review**
>
> Thank you for taking the time to read and review our paper.  We appreciate your feedback and below address questions you expressed.
>
> > What is the general recipe for applying this technique to get representations of a multiset?
>
> Sorry that this wasn't clear. The automaton is nondeterministic, so at each time step it could be in any state. The vector fw(w) of forward weights could be thought of as like a distribution over the state that the machine is in after reading w, except that the values don't have to be probabilities. This vector fw(w) is what we propose as the "general recipe" to represent multiset w.
>
> > It is not entirely clear what theoretical results are novel and which proofs are restatements of existing proofs.
>
> Again, sorry for not making this clear. Proposition 1 and Lemma 4 are not novel. To our knowledge, Propositions 2 and 3 are novel, as are the results in the appendices.
>
> > In what sense is the diagonal with alternating complex conjugate entries fully general?
>
> To be fully general, we should have put $r_j$ in front of each $\exp i\theta_j$ and $s_j$ in front of each $\exp i\phi_j$; apologies for this error. With that correction in place, the form at the top of page 5 is general in the sense that any real-weighted multiset automaton is close to a complex-weighted diagonal automaton (Prop 3), and because the original automaton had real weights, the diagonal entries that are complex must come in conjugate pairs. Those that are real can be duplicated to form conjugate pairs. Thus, any real-weighted multiset automaton is close to one that can be put into the form shown. We will try to make this clearer in a future version of the paper.
>
> > Since there are no confidence intervals it is impossible to draw conclusions from table 1.
>
> We've run bootstrap resampling to compare the other lines against the first line (the original position encodings). Roughly, significance is at about 0.4 BLEU. In the last line (learned per-position), all differences are significant except for Urdu-English. This confirms our conclusion that learned per-position encodings are worse, but the rest are all about the same.
>
> > The "units digit of a sum" task seems slightly artificially constructed to be suitable for a network which uses complex numbers. Although this is not a bad thing, it doesn't necessarily validate that complex weighted automata have better representational power. If that was the case, wouldn't we expect better results for other tasks that don't explicitly have a cyclic nature?
>
> The sum task demonstrates that DeepSets and our method are able to outperform LSTM and GRU models on multiset structured input, specifically being able to generalize results to multisets which are larger than were seen at training time.  The units-digit-of-sum task is meant to be a simple extension of the sum task to demonstrate that our method can not only represent the same types of data as the DeepSets method, but also represent other behavior such as cycles.  We have not run other tasks which don't explicitly have a cyclic nature for which DeepSets obtains less than 100% accuracy.

---

### Decision · Program_Chairs · 2019-12-19

**Decision:**

Reject

**Comment:**

Main summary: Paper is about generating feature representations for set elements using weighted multiset automata

Discussion:
reviewer 1: paper is well written but experimental results are not convincing
reviewer 2: well written but weak motivation
reviewer 3: well written but reviewer has some questions around the motivation of weighted automata machinery.
Recommendation: all the reviewers agree its well written but the paper could be stronger with motivation and experiments, all reviewers agree. I vote Reject.